# Q Fever in the First Trimester: A Case Report from Northern Rural New South Wales

**DOI:** 10.3390/tropicalmed4020090

**Published:** 2019-06-07

**Authors:** Sarah Marks, Maxwell Olenski

**Affiliations:** 1GP Obstetrician, Kempsey District Hospital, Kempsey, NSW 2440, Australia; 2Infectious Diseases Department, St Vincent’s Health, Melbourne, VIC 3065, Australia; Maxwell.OLENSKI@svha.org.au

**Keywords:** Q fever, pregnancy, antenatal screening, miscarriage, stillbirth, intrauterine growth restriction, fetal death in utero, preterm birth, first-trimester screening

## Abstract

Pregnant women are significantly more likely to have an asymptomatic acute infection with *C. burnetii* which, untreated, has been associated with poor obstetric outcomes including miscarriage, stillbirth, intrauterine growth restriction, and premature delivery. As such, Q fever is a potentially under-recognised and treatable cause of adverse pregnancy outcomes in rural Northern New South Wales, with testing of Q fever polymerase chain reaction (PCR)—whether on maternal sera or placental tissue—not currently recommended by the Perinatal Society of Australia and New Zealand for Stillbirth.

## 1. Introduction

Q fever, or ‘Query Fever’, is Australia’s most common and costly zoonosis [1]. Caused by infection with *Coxiella burnetii (C. Burnetii),* an obligate intracellular bacterium, 90% of Australian cases occur in northern New South Wales (NSW) and southern Queensland (QLD) (Figure A1). Livestock constitutes the main animal reservoir, and infected airborne particulates (predominantly from placental tissue, faeces, and urine) can give rise to outbreaks over vast areas. Whilst the incidence of symptomatic Q fever in Australia dropped initially with the introduction of a live, whole-cell vaccine for high-risk occupational exposures in both QLD and NSW in 1989, acute Q fever rates have risen again in more recent years, with reservoirs now extending beyond livestock to include kangaroo, possum, dingo, cat, fox, and wild pig [2]. The true incidence of Q fever rates in these areas is furthermore likely both under-recognised and under-reported, as an acute infection can be asymptomatic in as many as 80 percent of cases [3]. 

Moreover, pregnant women are more likely to be asymptomatic with acute infection and, therefore, may go undiagnosed [4]. This puts them at significant risk of chronic (or persistent) infection. Furthermore, untreated Q fever—particularly in the first trimester—has been associated with poor obstetric outcomes including early miscarriage, intrauterine growth restriction [IUGR], preterm birth, and foetal death in utero [FDIU] [5]. We report a case of Q fever in a pregnant woman during the first trimester in northern rural NSW, which serves to highlight some of the key diagnostic dilemmas associated with Q fever in pregnancy, and raises questions surrounding screening practices and pre-emptive treatment in seropositive pregnant women in rural Australia.

## 2. Case Discussion

The patient, a 39-year-old gravida 11 para 10 woman, presented to the local emergency department, seven weeks into her eleventh pregnancy with a three-day history of fevers, non-bilious vomiting, and right iliac fossa pain. Her bloods showed a mild elevation in liver transaminases, a mild lymphopenia and a moderately elevated C-reactive protein. Bedside ultrasonography revealed a viable intrauterine pregnancy with a small perigestational bleed. Her fever and abdominal pain subsided in the short stay observation unit, and she was discharged with antiemetic medication and follow-up with her general practitioner.

Apart from a past history of genital herpes simplex 2 infection, the patient disclosed no significant medical nor surgical history. She did not take any regularly prescribed medications beyond antenatal supplements, and had a documented anaphylactic reaction to penicillin.

Together with her husband and children, the patient resided on a large property reliant on tank water remote from the district hospital. The property was home to various animals, including cows, horses, chickens, ducks, dogs, and cats. All family members were well at the time of her presentation. She reported no recent travel, alteration in diet, neither animal nor insect bites, and the animals remained well on the property. She denied any illicit drug use, was a non-smoker, and drank no alcohol.

Two days following her initial presentation, the patient re-presented to the emergency department with ongoing vomiting and right iliac fossa pain. She described recurrent fevers and rigours at home. On this presentation, her bloods showed worsening liver dysfunction, lymphopenia and ongoing elevation in her inflammatory markers, with a new modest. 

She was admitted to the general surgical unit and commenced empirically on clindamycin for presumed acute appendicitis. Subsequent abdominal ultrasonography, however, showed no radiological signs of appendicitis and confirmed a viable intrauterine pregnancy. In light of her elevated liver transaminases, a liver panel was undertaken which included serology for Epstein-Barr virus and cytomegalovirus, as well as anti-neutrophil and smooth muscle antibody testing and serum lactate dehydrogenase, all of which were unremarkable. Atypical infectious investigations were also ordered, which included Q fever and arboviral serology. 

Over the course of her admission, her nausea and vomiting were managed with antiemetic medication, and her fevers and abdominal pain abated spontaneously. She was subsequently discharged after a three-day admission.

Following discharge, at nine weeks gestation, the patient was found to have a positive Q fever phase II IgG and IgM on indirect immunofluorescence assay (IFA) with titres of 3200 and >400, respectively (Table A1). She was contacted and commenced on cotrimoxazole therapy with high-dose folic acid. Follow-up investigations included transthoracic echocardiography, which demonstrated normal heart valves, left ventricle size, and systolic function. Both her liver function abnormalities and thrombocytopenia normalised over the ensuing month. 

The remainder of the patient’s pregnancy progressed uneventfully. She declined suppressive acyclovir in the third trimester and remained on twice daily cotrimoxazole until 36 weeks gestation. Repeat Q fever IFA serology at this stage remained strongly positive, with a phase I IgG of 800 and phase II IgG of 1600. Q fever PCR on serum was negative. 

She presented in spontaneous labour at 40 + 2 weeks gestation with membranes intact. Strict barrier nursing precautions were observed. On admission, the patient was examined and found to have no evidence of herpetic lesions. Her group B streptococcus status was unknown. She made good progress and spontaneously delivered a vigorous, live infant male weighing 3600 g. The third stage was actively managed with prompt delivery of the placenta, which showed no gross evidence of placentitis. The placenta was sent fresh for histopathology, with a small sample sent separately for Q fever PCR. Both PCR and histopathology were negative for Q fever placental infection, as too was breastmilk PCR. A blood sample was also taken from the baby for PCR, which was also negative.

The patient recommenced cotrimoxazole postpartum. Repeat Q fever IFA serology at three months indicated persistent infection, with Phase I IgG of 1600 and Phase II IgG of 100. At the time of writing, she was planned for repeat outpatient echocardiography and consideration of positron emission tomography to look for an infective focus. Ongoing surveillance serology was planned at three-monthly intervals, and the baby’s serology at three months remained negative.

## 3. Discussion

Diagnosing Q fever in symptomatic individuals requires a high index of suspicion from healthcare professionals, as the presenting cluster of symptoms is protean, and easily mimics a typical, albeit severe, flu-type illness. Pneumonia and hepatitis are also common presentations and, in the presence of fever, should prompt the physician to consider Q fever amongst the differential diagnosis, particularly in high-risk areas. Interpreting serology requires knowledge of the specific kinetics of antibody responses to phasic antigens to distinguish between acute and chronic infection, and can incorporate PCR-based testing (Table A2, Figure A2) [6]. Culture-based methods require immune staff and physical containment level 3 [PC3] laboratories owing to Q fever’s highly infectious nature [5].

Pregnant women are significantly more likely to have asymptomatic acute infection with *C. burnetii*. Untreated, Q fever has been associated with poor obstetric outcomes including early miscarriage, IUGR, preterm birth, and FDIU [7]. No direct causal relationship between Q fever and these outcomes has been demonstrated to date; however, there is a growing body of evidence that seroconversion in the first trimester confers a higher risk of sequelae. Early miscarriage and stillbirth also appear to be associated with placental infection [8,9]. 

One study of 74 women from India reported an association of serologically confirmed acute infection with first-trimester miscarriage at a rate greater than 25%. The study utilised both PCR and immunofluorescence assay on pregnancy tissue, maternal sera and genital swabs, and strengthened the supposition that prevalence of Q fever in vulnerable communities may be grossly under-recognised [10]. 

In addition to the higher likelihood of asymptomatic infection in pregnancy, women are more likely to develop a persistent—or chronic—manifestation of infection, which confers a high prevalence of complications including endocarditis. Pregnant patients presenting with acute Q fever should have transthoracic echocardiography performed to assess for the presence of pre-existing valvular disease, which heightens the risk of subsequent endocarditis and chronic infection [11]. 

The safest and most effective treatment of Q fever in pregnancy appears to be twice daily co-formulated trimethoprim with sulfamethoxazole (cotrimoxazole), marketed in Australia as Bactrim [7]. Owing to folinic acid antagonism, this combination is not without risk in pregnancy in the first trimester during organogenesis and high dose folate is an important addition to treatment regimens. Ceasing treatment at 36 weeks is thought to reduce the risks of kernicterus to the newborn [12]. Carcopino et al. found that the treatment with cotrimoxazole beyond five weeks reduced the incidence of placentitis, adverse obstetrics outcomes including stillbirth, and the incidence of persistent maternal infection [13].

The feasibility of a screening program for pregnant women is an area of much-needed study. During a major outbreak, one clustered randomised control trial undertaken in The Netherlands found no statistically significant improvement in obstetric outcomes with long-term therapy with either cotrimoxazole or erythromycin in cases of serologically confirmed acute Q fever from 20 weeks gestation [14]. While this represents the only randomised control trial to date, major limitations include the lack of statistical power, the use of macrolides which are thought to be less effective than cotrimoxazole in treating Q fever, and a paucity of first-trimester pregnancies included in the study population, which represent the target population for the intervention [11]. The ethics of screening and treating a population that may fall prey to sequelae from either treatment or a positive screening result is a major obstacle that will inform future research efforts. Moreover, the fact that a direct causal relationship between laboratory-confirmed Q fever and adverse pregnancy outcomes is inconsistent across the current literature which also suggests varying virulence and pathogenicity amongst the *C. burnetii* clades, which tend to be geographically distinct. 

## 4. Conclusions

Q fever is a potentially under-recognised and treatable cause of miscarriage and late pregnancy loss in rural Northern NSW, for which further study is warranted. At the time of writing, we note that Q fever PCR—whether on maternal sera or placental tissue—are not included in the current investigations recommended by the Perinatal Society of Australia and New Zealand for Stillbirth. Whether or not to treat, particularly in asymptomatic first-trimester pregnancies, remains contentious and the subject of future research direction. Finally, without a high index of suspicion and familiarity with the infection’s manifestations, Q fever diagnosis is sometimes missed.

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
