# Peer review of "Q Fever in the First Trimester: A Case Report from Northern Rural New South Wales"

_tropicalmed, 2019, doi:10.3390/tropicalmed4020090_

Round 1
Reviewer 1 Report
Thank you for the opportunity to review this case report.
This is a succinctly written case report outlining a case of diagnosed Q fever in rural New South Wales Australia; a region with high notification rates of Q fever. Given this latter fact, Q fever-associated reports of obstetric complications and treatment regimes out of this country are rare so this article will have interest to Australians practitioners and could have interest internationally given the inconsistent findings and outcomes associated with Coxiella burnetii infection worldwide which requires further investigation.
The case report is well written however below are a few minor suggestions for improvement:
Generally, throughout the manuscript Q Fever should be changed to Q fever (lower case f)
Lines 24-26 – the sentence beginning “While the incidence of symptomatic Q fever has dropped significantly since the introduction of a live whole cell vaccine….”. This was true initially however it is worth noting that the incidence has risen again in recent years with >600 notifications in 2015 with a notable rise in the number of notifications not associated with traditional reservoirs such as livestock. This is important from the perspective of this paper as the index of suspicion will be even lower in such cases.
Line 28 – asymptomatic infections ranges from 20-80% depending on geographic region (Reference: Million M, Raoult D. Recent advances in the study of Q fever epidemiology, diagnosis and management. Journal of Infection. 2015;71:S2-S9).
Lines 69-70, 77-78, 88-89 and Table A1 – the authors should make it clear that they are referring to indirect immunofluorescence (IFA) in regard to these titres.
Line 86 – change “taken FOR the baby” to “taken FROM the baby”.
Line 105 – misspelling burnetii - one t
Line 139-141 – might as well quote reference 9 in support of this statement as this is the conclusion the authors of this paper also come to given the discrepancy between the pregnancy outcomes in their study compared to the outcomes in The Netherlands (reference 12).
Line 165 – Reference 1 – incorrect author listing – Katrina B. should read Bosward K.
Author Response
Thank you for your valuable feedback.
We have amended the basic spelling/reference errors and considered your valid point that Q fever rates are indeed on the rise in northern NSW/ southern QLD.
We feel a potential study looking at the placental tissue of early miscarriage and still birth (perhaps with Q fever PCR) in this region would be an interesting and ethical way of attempting to gauge the incidence of Q fever in women with pregnancy loss compared to its point prevalence in the general population.
We hope that the article generates some discussion amongst health practitioners both in the obstetrics and primary care fields in rural medicine and heightens awareness amongst clinicians to consider this protean infection as a diagnosis in the obstetric population.
Reviewer 2 Report
The discussion concerning the possibility of screening stillbirths for Coxiella is interesting and warrants further consideration.
It would be useful to add in the numerical biochemistry and haematology results for the case study, as well as to state which particular Q fever serology kit was used as the titres obtained are highly dependent on this.
Author Response
Thank you for your review. We have updated the manuscript to include the kit used to test the patients' serology (micro immunofluoroescence). Incidentally, the kit is generated in house by the Australian Rickettsial Reference Lab (Geelong).